# Hydrothermal Synthesis and Properties of Nanostructured Silica Containing Lanthanide Type Ln–SiO_2_ (Ln = La, Ce, Pr, Nd, Eu, Gd, Dy, Yb, Lu)

**DOI:** 10.3390/nano13030382

**Published:** 2023-01-18

**Authors:** Joana M. F. Barros, Glauber J. T. Fernandes, Marcio D. S. Araujo, Dulce M. A. Melo, Amanda D. Gondim, Valter J. Fernandes, Antonio S. Araujo

**Affiliations:** 1Center of Education and Health, Academic Unit of Biology and Chemistry, Federal University of Campina Grande, Cuite 58175-000, PB, Brazil; 2Laboratory of Catalysis and Petrochemistry, Institute of Chemistry, Federal University of Rio Grande do Norte, Natal 59078-970, RN, Brazil; 3Institute of Chemistry, Federal University of Rio Grande do Norte, Natal 59078-970, RN, Brazil; 4Laboratory of Fuels and Lubricants, Institute of Chemistry, Federal University of Rio Grande do Norte, Natal 59078-970, RN, Brazil

**Keywords:** lanthanides, silica, hydrothermal synthesis, nanostructured materials

## Abstract

The nanostructured lanthanide-silica materials of the Ln–SiO_2_ type (Ln = La, Ce, Pr, Nd, Eu, Gd, Dy, Yb, Lu) were synthesized by the hydrothermal method at 100 °C, using cetyltrimethylammonium as a structural template, silica gel and sodium silicate as a source of silicon, and lanthanide oxides, with Si/Ln molar ratio = 50. The resulting materials were calcined at 500 °C using nitrogen and air, and characterized by X-ray diffraction (XRD), Fourier-Transform infrared absorption spectroscopy, scanning electron microscopy, thermogravimetry (TG), surface area by the BET method and acidity measurements by n-butylamine adsorption. The XRD and chemical analysis indicated that the SiO_2_ presented a hexagonal structure and the incorporation of lanthanides in the structure changes the properties of the Ln–SiO_2_ materials. The heavier the lanthanide element, the higher the Si/Ln ratio. The TG curves showed that the decomposition of the structural template occurs in the materials at temperatures below 500 °C. The samples showed variations in specific surface area, mean pore diameter and silica wall thickness, depending on the nature of the lanthanide. The incorporation of different lanthanides in the silica generated acid sites of varied strength. The hydrothermal stability of the Ln–SiO_2_ materials evaluated at high temperatures, evidenced that the properties can be controlled for application in adsorption and catalysis processes.

## 1. Introduction

The discovery of silica-based nanostructured materials opened new perspectives for the development of new materials with organized structures containing heteroatoms [1,2]. Due to the high surface area and accessibility of their pore systems, these materials have been promising as acid catalysts in petrochemical processes [3,4,5], supports for heteropolyacids [6,7,8], liquid phase catalysis [9,10,11], and in advanced materials technology [12,13,14,15,16,17]. Due to the great interest in these materials, it is necessary to develop new methodologies for synthesis, post-synthesis treatments, in addition to new characterization methods. The silica-based Mesoporous Composition of Matter number forty one (MCM-41) is one of the most important materials developed so far with hexagonal arrangement of one-dimensional mesopores with diameters ranging from 2 to 10 nm, good thermal stability, high area values specific and pore volume. These characteristics have made the MCM-41 a promising material for applications in catalysis, adsorption and in the technology of advanced materials based on molecular sieves, such as: electron transfer photosensors, semiconductors, polymers, carbon fibers, clusters, and materials with non-linear optical properties [17,18,19,20,21,22].

Recently, lanthanides have been applied in several areas of interest [23,24,25,26,27]. They are the most active elements in catalysts for oxidative coupling of methane to form higher hydrocarbons. Cerium oxide has been used as a catalyst or structural promoter for metallic oxides. Structural promotion is attributed to the capacity of the lanthanides to form crystalline oxides with reticular defects which can act as active sites [28]. The effect of this promotion is given by the excellent thermal and mechanical resistance of the lanthanide oxide in the catalyst surface, as well as acidity of the catalysts [29].

The study of acid solids, such as Silica-MCM-41, for application as catalysts or catalytic supports, has been previously reported [30,31]. Acidity is considered an ionic property of catalysts, responsible for their activity in converting hydrocarbons, playing important roles in organic reactions that occur on solid catalysts, mainly in the petroleum and petrochemical industries. Determining the strength of the acidic sites exposed on the surface of the solid, as well as their distribution, are fundamental conditions for preview their activity and selectivity, in order to allow relating the catalytic properties with the acidic ones. The main properties that the acid sites of a catalyst must present are nature (Brönsted and Lewis sites), strength (temperature range) and density (total acidity). There are a variety of methods to determine the surface acidity of these solids, the difference of each consisting of physical and chemical principles. The Temperature Program Desorption (TPD) of base molecule probe adsorbed on the catalyst have also been used for this monitoring, using thermogravimetry [32,33].

Several works have reported an increase in the thermal stability of zeolites [32] and other materials [34] after the addition of lanthanide cations, mainly those belonging to the lighter group (La, Ce, Nd, Pr). The role of La and Ce cations in the thermal stability of zeolites is already well known [32,33]. However, the rare earth application market has changed direction, motivated by the extensive application of these elements in the high technology industry. Due to this scenario, the study of the properties of lanthanide incorporated into silica materials has been of great importance [34]. The thermal and hydrothermal stability of nanostructured materials such as MCM-41 [35,36] and MSU-X [37] can be increased by incorporating a small amount of lanthanide elements. The knowledge of the characteristics of lanthanide elements has stimulated several researchers to investigate new synthesis routes and chemical formulations for the development and applications of these materials in adsorption and catalysis [38,39,40]. Lanthanum can be used efficiently as compensation cation in perovskite structures, in order to obtain nanostructures such as Solid Oxide Fuel Cells (SOFCs) for energy storage [41]. 

In this work, the incorporation of lanthanide (Ln) elements in the SiO_2_-based nanostructured material was investigated. The Ln cations were introduced into the reaction medium in the form of their respective oxides. With the objective of verifying how the Ln present in the gel affects the properties of the final material, samples of Ln–SiO_2_ were synthesized, where Ln = La, Ce, Pr, Nd, Eu, Gd, Dy, Yb and Lu. The synthesized materials were characterized for the evaluation of crystallographic, structural, and morphological properties, and acidity. These properties are important for catalytic and adsorption applications.

## 2. Materials and Methods

### 2.1. Precursors and Reagents

The precursors used for the synthesis of the materials were silica gel 95% SiO_2_ and 5% H_2_O from Merck (Rahway, NJ, USA), and sodium silicate 63% SiO_2_ e 18% Na_2_O from Riedel Der Häen (Hanover, Germany). The structural template used was cetyltrimethylammonium bromide (CTMABr) 98%, from Vetec Quimica (Recife, PE, Brazil). The Ln elements (La, Nd, Nd, Eu, Gd, Dy, Yb, Lu) oxides were 99%, purchased from Sigma Aldrich (St. Louis, MO, USA). The praseodymium oxide (99.9%) was purchased from Alpha Products (Bedford, IL, USA). For Ce, was used cerium chloride (96%). The reactants used as solvents were distilled water, hydrochloric acid 37% and ethanol 95%, from Merck. A solution of 30% vol. of acetic acid solution was used for pH adjustment, and sodium acetate was used for stabilizing the materials.

### 2.2. Hydrothermal Synthesis Procedures

The materials were synthesized by the hydrothermal method using silica gel, sodium silicate, lanthanide source, structural template, and water. These reagents were added in stoichiometric proportions in order to obtain a gel with molar composition: 1.0 CTMABr x Ln_2_O_3_: 4.0 SiO_2_: 1.0Na_2_O: 200 H_2_O. (the “x” value was adjusted so that the Si/Ln molar ratio remained equal to 50 in the samples). The experimental procedure is represented schematically in Figure 1.

For the synthesis, a mixture prepared from the silicon oxide, sodium silicate and lanthanide source were added to half of the water required for the synthesis. The system was stirred for 2 h at 60 °C, then an aqueous solution of the template (CTMABr plus the other half of the water) was added to the mixture, which remained under stirring at room temperature. After preparing the gel, it was transferred to a Teflon vessel and placed in a stainless steel autoclave and heated in an oven at 100 °C for 120 h (5 days), every 24 h the pH of the gel was adjusted to the range between 9.5 to 10.0 with a 30% solution of acetic acid, and when the pH stabilized, sodium acetate was added in a molar ratio CTMABr/CH_3_COONa = 3 for complete stabilization of the silica. Then the system was remained for another 24 h at 100 °C. After crystallization, the autoclave was removed from the oven and cooled to room temperature. Its contents were washed with distilled water and then with a 2% solution of hydrochloric acid in ethanol to remove the surfactant. The solid resulting from the crystallization process was dried in at 100 °C for 12 h. For comparation, the material without Ln was synthesized according to the same procedure described without addition of the lanthanides in the gel composition.

According to Figure 1, the last step to obtain the nanoporous materials is the calcination. The calcination process of the series of the SiO_2_ and Ln–SiO_2_ nanomaterials occurred in two steps where initially the samples were subjected to a heating hate of 10 °C/min from room temperature to 500 °C in inert nitrogen atmosphere flowing at 100 mL/min. After reaching the temperature of 500 °C, the system remained under this condition for 1 h. Then, the nitrogen flow was changed for synthetic air at the same flow rate for an additional time of 1 h. This calcination process aims to remove the structural template from the pores of the materials.

### 2.3. Characterization of the Materials

The synthesized catalysts in the form of oxides were submitted to chemical analysis via X-ray fluorescence (XRF) by dispersive energy in a Shimadzu equipment model EDX-800 (Kyoto, Japan), where the concentrations of Si, Y, La, Ce, Nd, Pr, Ga, Eu, Dy, Yb and Lu.

The characterization by X-ray diffraction (XRD) was carried out in a Shimadzu equipment, model XRD-6000 (Kyoto, Japan), using a CuKa radiation source with voltage of 30 kV and current of 30 mA, with Ni filter. The data were collected in the 2 q range from 1 to 10 degrees with a goniometer speed of 2 °/min with a step of 0.02 degrees. 

Absorption spectra in the infrared region were obtained in a Bomem model MB 102 Fourier transform infrared spectrophotometer, using the KBr method, and ca. 1% of material dispersed. The spectra were obtained in the wavenumber region of 4000–400 cm^−1^. 

The thermogravimetric analyzes (TG/DTG) of the as synthesized materials were carried out in a Mettler Toledo TGA/SDTA 851 thermobalance, at a heating rate of 10 °C/min, in temperature range of room temperature up to 900 °C, using nitrogen atmosphere at flow rate of 25 mL/min. In all analyses, 70 mL alumina crucibles and a sample mass of approximately 15 mg were used. From the thermogravimetric curves of the samples before the calcination stage, the amounts of water and organic template (CTMABr) were determined, as well as the temperature ranges where these molecules were removed. These data were essential for determining the lowest calcination temperature necessary to remove the organic template from the pores of the materials.

The microscopies of the samples were obtained using a Philips scanning electron microscope, model ESEM. The procedure for preparing the materials for analysis consisted of depositing a portion of the solid on an adhesive carbon tape attached to the sample holder. Then, a thin layer of gold was deposited to improve sample conduction. Micrographs were obtained at magnifications ranging from 500 to 2000 times.

The surface area was determined through the adsorption of N_2_ at 77 K using the BET method in a Quantachrome equipment model NOVA-2000. Before each analysis, approximately 0.5 g of the previously calcined sample, was treated at 200 °C under vacuum for 3 h. This treatment aims to remove moisture from the surface of the solid. The N_2_ adsorption isotherms for the samples were obtained in the P/Po range between 0.1 and 0.9, allowing important information to be obtained about the materials, such as: surface area, external area and mesoporous volume. Surface area data are important to evaluate the influence of lanthanide incorporation in the mesoporous SiO_2_ materials.

### 2.4. Surface Acidity Measurements

The total acidity of the mesoporous materials obtained was determined by the base adsorption method followed by desorption by temperature increase carried out in a thermogravimetric balance model Mettler Toledo TGA/SDTA 851. The base used as a probe molecule was n-butylamine.

The procedure consisted of initially heating the already calcined samples from room temperature to 400 °C, for sample activation, with a N_2_ flow of 30 mL/min for 40 min. After this period, the temperature was reduced to 95 °C and the n-butylamine vapors were continuously directed to the sample through the N_2_ flow for 1 h, for complete saturation of the acid sites present in the material. Then, the saturated samples were purged with N_2_ at the same saturation temperature, for 30 min, to remove the physically adsorbed base. Figure 2 shows the adsorption system used to study the acidic properties of materials.

After the treatment for saturation of the sample with the probe molecule, the thermodesorption of n-butylamine was carried out in a thermobalance at a heating rate of 10 °C/min, from room temperature to 900 °C, under a nitrogen flow of 25 mL/min. Total acidity was calculated as a function of the amount of thermodesorbed n-butylamine in each sample according to Equation (1).
(1)$$A=\frac{1000\cdot m\cdot M_{0}}{M\cdot M_{F}}$$
where: A = acidity (mmol/g); m = mass loss in the event; M_0_ = initial mass of the sample (g); M_F_ = final mass of the sample (g); M = molecular mass of n-butylamine (73 g/mol).

## 3. Results

### 3.1. Chemical Composition from XRF

The chemical compositions of pure SiO_2_ and Ln–SiO_2_ samples are shown in Table 1. These data show the presence of lanthanide ions in all synthesized samples, proving the efficiency of the synthesis method. The energy dispersive X-ray fluorescence (XRF) spectra of the catalysts are shown in Figure 3. From this analysis, the concentrations of the lanthanide elements present in the samples were determined.

According to the Figure 3, four regions are observed in the spectra, corresponding to lines from energy dispersion. The peaks in the regions (i) and (iv) are typical of silica; whereas the regions (ii) and (iii) are indicative of the presence of lanthanides in the silica structure. The peaks shift from La to Lu, due to number of electrons in the 4 f orbitals. In the compounds La–SiO_2_, Pr–SiO_2_, Nd–SiO_2_ and Eu–SiO_2_, only peaks in the region (ii) were observed. The presence of peaks in region (ii) are probably due to a high degree of covalent coordination of the Ln–O–Si bonds (Ln = La, Pr, Nd and Eu) in the materials, suggesting the presence of trivalent cations of the lanthanides [42,43]. For Ce–SiO_2_, Gd–SiO_2_, Dy–SiO_2_ and Lu–SiO_2_, high intensity peaks were observed in the region (iii), probably to a mix valence of cerium, forming Ce_3_O_4_ by the combination of CeO with Ce_2_O_3_, and presence of heavy elements occupying different positions in the silica structure, in the case of Gd, Dy and Lu, with subsequent increasing in the Si/Ln ration (see Table 1).

According to the results shown in Table 1, it is observed that the amount (%) of lanthanides incorporated in the mesoporous material after the calcination step is lower than the amount present in the synthesis gel in all samples. This may be related to some characteristic properties of lanthanides, such as: electronegativity, lanthanide contraction, and hydration during the hydrothermal synthesis. The experimental correlations obtained between the percentage of incorporated lanthanides and the ionic radius, electronegativity, and degree of hydration of the studied elements are presented in Figure 4, Figure 5 and Figure 6, respectively.

According to Table 1 and Figure 4, Figure 5 and Figure 6, it is observed that the percentage of lanthanide impregnated in the silica obeys the following order: Eu > La > Gd > Ce > Pr > Nd > Dy > Yb > Lu. With the exception of Eu and Gd, the percentage of impregnated lanthanide decreases with the contraction of the lanthanide series and increases as the electronegativity and hydration energy. In the presented correlations, the samples containing europium and gadolinium are presented in a discordant way, this must be due to a particular property that is conferred to these elements, which is the high stability attributed to them due to the half-filling of the 4f orbitals. For lanthanum, as there are no 4f electrons, there is no interelectronic repulsion, consequently it presents greater stability in the silica structure. For the other elements, an electrostatic effect is observed directly associated with the increase in the nuclear charge imperfectly shielded by the 4f electrons.

### 3.2. Crystallographic Properties from XRD

The X-ray diffractograms of the mesoporous material SiO_2_ and Ln-SiO_2_ are shown in the Figure 7 with the presence of four Bragg angles that are represented by the reflections of the planes (100), (110), (200) and (210) [2]. 

The existence of a single plane (100) has been determinant to evidence the presence of the hexagonal mesopore structure, like that observed to SiO_2_–MCM-41 [2]. Thus, the mesoporous parameter (a_0_) of the materials structure is obtained through the reflection peak for the (100) plane, which is the most characteristic in the X-ray diffractogram, and is calculated by the following equations [2]:(2)$$\frac{1}{{d_{\left(\mathrm{hkl}\right)}}^{2}}=\frac{4\;\left(h^{2}+\mathrm{hk}+l^{2}\right)}{3\;{a_{0}}^{2}}+\frac{l^{2}}{c}$$
(3)$$\frac{1}{{d_{\left(100\right)}}^{2}}=\frac{4}{3\;{a_{0}}^{2}}$$
(4)$$a_{0}=\frac{{2\;d}_{\left(100\right)}}{\sqrt{3}}$$
where: d_(hkl)_ = interplanar distance relative for the (100) plane; a_0_ = mesoporous parameter of the hexagonal array of the SiO_2_ structure.

The interplanar distance for the plane (100) is calculated by the Equation (5):(5)$$\lambda_{\mathrm{CuK}\alpha }={2\;d}_{\left(100\right)}\;\mathrm{sen}\;\theta$$
where: CuKa = wavelength for CuKa = 1.5418 nm.

A peak at 2 q = 2.19 nm corresponding to the reflection (100) in the hexagonal SiO_2_ sample, gives us according to Bragg’s rule (Equation (5)), the dimension of the hexagonal lattice parameter (a_0_) of ca. 4.65 nm. For the samples containing lanthanides, a decreasing in a_0_ was observed, indicating that there was a contraction of the lattice. In samples containing La, Ce, Pr, Nd, Eu and Gd, the contraction of the structure was not followed by a decreasing in the structural order.

The decreasing in the reflection intensity of planes (100, 110 and 200) in samples Dy-SiO_2_, Yb–SiO_2_ and Lu–SiO_2_, indicate a disordering of the hexagonal structure, suggesting an imperfection in the channels. This can be explained by the difficulty of incorporating the heavier lanthanides into the SiO_2_ network, since the size of trivalent lanthanide ions is a determining property for the stability of their coordination with oxygen. Closed packings of oxygen with silicon and lanthanide cations accommodated in octahedral and tetrahedral structures are not found in silicates containing lanthanides where the contraction is more pronounced. The anionic part of the structure, represented by oxygens and tetrahedral [SiO_4_] can be adapted to lanthanide cations until reaching the maximum degree of distortion allowed in the tetrahedral structure of the individual silicate. The electrostatic character of the bond and the strongly polarizing forces of the heavier lanthanide ions against oxygen result in extreme distortion of [SiO_4_] tetrahedra [40]. These results are summarized in Table 2 and indicate that obtaining the ordered SiO_2_ phase is possible with the incorporation of the lighter lanthanides in the structure.

### 3.3. Structural Properties from FTIR

The infrared absorption spectra in the wavenumber range of 4000–400 cm^−1^ for SiO_2_ and Ln–SiO_2_ materials are shown in Figure 8. Infrared spectra have been widely applied for the characterization of silica for characterization of the vibration bands of the asymmetric and symmetric Si–O stretching present in the structure, which appear in the wavenumbers of 1243–1091 cm^−1^ and 806 cm^−1^, respectively. Infrared spectra have been widely applied for the characterization of silica for characterization of the vibration bands of the asymmetric and symmetric Si–O stretching present in the structure.

The spectra were carried out before and after calcination, noted in the Figure 8 as letters (a) and (b), respectively. In this figure was selected samples for SiO_2_, and samples with light lanthanides La–SiO_2_ and Eu–SiO_2_, and heavy lanthanides Yb–SiO_2_ and Lu–SiO_2_. The infrared spectra for the samples show five regions: (i) a broad absorption band centered at 3440 cm^−1^, corresponding to the O-H stretching bands of water molecules and the Si–O–H stretching of silanol groups bonded to water [43]. 

The calcination process removed the template, increasing the hygroscopicity of the materials, and consequently the water physisorption. The region (ii) is attributed to cethyltrimethylammonium (CTMA+) ions still remaining in the materials, with two bands at 2910 and 2815 cm^−1^ due to asymmetric and symmetric stretching vibrations of the alkyl chain, respectively.

The region (iii) shows a vibration band at 1640 cm^−1^ evidencing the presence of water. A weak absorption band at ca. 1480 cm^−1^ corresponds to bending vibration of the C–H groups of the CTMA+ template, and consequently they are not observed after calcination. The broad absorption bands in region (iv) centered at 1055–1085 cm^−1^, 785–805 cm^−1^ and 450–455 cm^−1^ in region (v) were attributed to asymmetric stretching, symmetric stretching and bending vibration of the Si-O-Si network, respectively [44].

In general, the uncalcination samples show absorption bands at ca. 2930 com^−1^ and 2800 cm^−1^, corresponding to the template species still remaining inside the pores of the materials, whereas for the calcined samples, the absence of this band indicates the complete removal of the organic template from the original structures. The vibrations related to deformations of adsorbed molecules cause the appearance of bands at 1640 cm^−1^ for SiO_2_ and 1640 cm^−1^ and 1480 cm^−1^ for the samples containing La, Pr, Nd and Eu. For the samples containing Ce, Gd, Yb, Dy and Lu, additional bands in the regions of 1568 cm^−1^ and 1405 cm^−1^ were observed.

The spectra of the samples containing La, Ce, Pr, Nd, Eu and Gd, present a band in the region 960 cm^−1^ less intense than in the pure SiO_2_ sample, that can be explained by the differences caused in the structure containing heteroatoms. The presence of vibrations related to stretching of Si–OH groups present in samples suggest the presence of structural defects caused by the incorporation of other elements in the lattice of SiO_2_. The disappearance of this band in the spectra of samples containing Dy, Yb and Lu evidence the imperfect incorporation of these elements in the SiO_2_ structure, as already verified from XDR and XRF, where the effect of lanthanide contraction is strongest.

### 3.4. Scanning Electron Microscopy

The morphology of the samples was characterized by scanning electron microscopy as shown in Figure 9. The obtained results show that pure SiO_2_ is formed by the agglomeration of rounded particles. For the Ln–SiO_2_ materials, in general observed an agglomeration of small particles of different shape and sizes is observed, due to interaction of Si–O–Ln in the structure of the materials, and some lanthanum oxides impregnated on the surface.

### 3.5. Thermal Analysis

Thermogravimetric analyzes of the uncalcined samples, obtained in a nitrogen atmosphere, is shown in Figure 10. The thermogravimetric analysis presents three main mass losses, signed as: (I) desorption of physically adsorbed water; (II) surfactant decomposition; (III) condensation of silanol groups. The Ce-SiO_2_ and Lu-SiO_2_ are representative for light and heavy lanthanides. Table 3 presents the temperature ranges and the respective mass losses for each event for all samples.

According to the data in Table 3, it is observed that in all samples there is no significant variation in the temperature range for the removal of water molecules, as well as for the decomposition of the surfactant. The difference in the percentage of mass loss between the materials relative to the first event is attributed to the humidity that each sample was exposed to before the thermogravimetric analysis, since it deals with physically adsorbed water and mesoporous silica-based materials [45,46]. There is also the possibility that it is related to intracrystalline water molecules, which could have a correlation between this percentage of mass loss with the hydration energy of the lanthanides, but since there is no possibility of quantifying these two events separately, this correlation becomes unavailable. Regarding the second stage, the difference in the mass percentage attributed to surfactant removal still remaining inside the pores of the materials even after previous washing with a solution of HCl in ethanol, suggesting that the pores must be stricter, making it difficult to remove cetyltrimethylammonium ions to the outer surface of the material. These results are in agreement with those presented by the absorption spectra in the infrared region where these samples that present a higher percentage of mass loss are the same ones that present more intense bands in the region of absorption of the surfactant molecules.

In the third stage of mass loss presented in the thermogravimetric analyses, the variation in the percentage of mass loss may be an indication that the rare earth incorporated into the synthesis gel of the SiO_2_ nanomaterial interferes with the condensation of the silanol groups, the bonds between oxygen, silicon and lanthanides occurs with different interactions. The samples containing lanthanum and europium were the ones that presented a smaller mass loss, indicating that there is condensation of the silanol groups (Si–OH and Ln–OH) located inside the pores of the structure and not on the silica walls, this causes this removal to be more difficult. The greater the mass loss related to the condensation of the silanol groups, the more likely the hypothesis that this process occurs in the silica walls of the materials, which makes the removal of this species possible in the temperature range between 350 and 540 °C. 

Samples containing Pr and Nd showed similar mass loss to the SiO_2_ material. Materials containing Ce, Gd, Dy, Yb and Lu showed a greater mass loss ranging from 5 to 7%. This result can be explained due to the non-stability of the bonds of the heavier lanthanides with the silica tetrahedrons, occurring instability of the ion, passing from Ce^+3^ to Ce^+4^ changing the properties of Ce–O bonds [47]. 

### 3.6. Surface Acidity Properties from TG

The thermogravimetric (TG/DTG) curves of the materials previously adsorbed with n-butylamine showing the thermal desorption of this base molecule from the acid sites of the materials are shown in Figure 11. 

The curves show two main events: The first (I) in the temperature range from 30 to 80 °C, corresponds to water desorption and physically adsorbed amine. The second (II), in the range of 80 to 330 °C attributed to the desorption of chemically adsorbed n-butylamine on the acidic sites of the materials. Data from the literature [48,49] on the adsorption of amines for the characterization of surface acidity indicate that desorption of n-butylamine molecules retained in acid sites (chemical adsorption) occurs through the decomposition of the ion formed by the protonation of n-butylamine by the acid site. This reaction can occur in two ways, similar to the Hoffman degradation, where the rupture of the C-N bond occurs producing ammonia and butene (main form of decomposition) or through the rupture of the C-C bond forming methylamine and propene. In the samples Ln-SiO_2_, the lanthanides were used as acid site promoters. These active sites could be generated by the deprotonation of surface lanthanide species [50].
[Ln(OH_2_)]^n+^ → [Ln(OH)]^(n−1)+^ + H^+^(6)

The protons formed during this process must interact with oxygen bound to the silicon and form acid silanol groups of the SiOH-H^+^ type, which increase the total acidity of the material. The silanol form Bronsted acid sites, while cationic hydroxyls generate Lewis acidity. The temperature ranges and total acidity, in mmol/g, is given in Table 4.

The samples containing La, Ce, Pr and Eu showed a higher surface acidity than the pure SiO_2_ material. The Eu–SiO_2_ sample showed a higher temperature range in the second stages of mass loss, probably due to the decomposition of the protonated amine by the acid site in narrow channels or by the resorption of the fragments decomposed, mainly ammonia and methylamine, which desorb only at higher temperatures.

### 3.7. Surface Area from BET

The Figure 12 shows the N_2_ adsorption/desorption isotherms at 77 K. All samples show type IV isotherms according to the IUPAC classification [51,52], with low adsorption at relative pressures <0.1. 

From the results of this analysis, it was possible to determine the specific surface area by the BET method, the mean pore diameter by the BJH method [53] and the silica wall thickness according to Equation (7) [54]. These data are shown in Table 5.
ω = a_0_ − d_p_(7)
where: ω = silica wall thickness; a_0_ = mesoporous parameter and d_p_ = pore diameter.

The adsorption data show that the amount of nitrogen adsorbed gradually increases with increasing the relative pressure by multilayer adsorption. The rapid increase in the adsorbed amount is observed over the P/P_0_ from 0.3 to 0.4 due to the condensation of nitrogen in the mesopores. For the pure SiO_2_ sample, the adsorption curve practically coincides with the desorption curve. Samples modified with lanthanides show a marked hysteresis at relative pressures 0.3 and 0.4, this may be indicative of different interactions of the nitrogen molecule in the pore structure. The samples containing Lu, Dy, Y and Yb show continuity in nitrogen adsorption at relative pressures greater than 0.4, this may be associated with the presence of larger pores in the structure of these materials.

The variation in the pore diameter of the samples containing lanthanides when compared to the pure SiO_2_ is attributed to the differences in the bond strengths between Si–O–Si and Ln–O–Si, as well as to the location of these interactions, that is, if the greater contribution is inside or outside the pores, forming structures with smaller pores and consequently with greater silica wall thickness or generating structures with larger pores with less resistant wall thickness. The chemical bond strength between Si–O is about 217 kJ/mol, while for Ln–O (Ln^3+^), it amounts to ca. 430–449 kJ/mol [47]. The heavier the lanthanide, the greater its chemical bonding strength. 

## 4. Conclusions

The results of the different physicochemical analyses allowed the following conclusions: the materials of the pure SiO_2_ and containing lanthanides (Ln–SiO_2_) were successfully synthesized through the hydrothermal method, using cetyltrimethylammonium bromide as a structural template. The lanthanide impregnated in the synthesis gel interferes with the properties of the resulting material. The amount of lanthanide present in the material after calcination is not the same in all samples. This variation occurs due to some characteristic properties of the lanthanide series. In general, the amount of impregnated lanthanide decreases with the contraction of the lanthanide series and increases as the electronegativity and hydration energy, with exceptions for europium and gadolinium, due to a particular property that is conferred to these elements, which is the high stability attributed to them due to the semi-filling of the 4f orbitals. The XRD patterns of all samples showed three typical peaks, one with a high intensity, attributed to the reflection line of the plane (100) and two others with lesser intensity attributed to the reflections of the planes (110) and (200) characteristic of the mesoporous hexagonal structure, similar to Mesoporous Composition of Matter (MCM-41). The samples Dy–SiO_2_, Yb–SiO_2_ and Lu–SiO_2_ showed a decrease in the intensity of reflection of the planes (100, 110 and 200) evidencing an irregular arrangement of pores and channels, due to the difficulty of incorporating the heavier lanthanides into the SiO_2_ network. The infrared absorption spectra of the materials SiO_2_ presented bands characteristic of asymmetric and symmetric Si–O–Si stretching, from the vibrations, Si–OH, and for Ln–SiO_2_, the band related to the Ln–Si–O vibrations is noticeable in samples containing the lightest rare earths and is practically not observed in samples modified by lanthanides with smaller ionic radius. Furthermore, through the FTIR spectra it is possible to verify that the calcination process removes all the organic material contained in the materials, since the spectra of the calcined samples do not present the band in the region related to the organic material. Thermogravimetric analyzes of the materials present the characteristic mass losses, the first being related to the output of physissorbed water on the surface, the second to the decomposition of the surfactant and the third related to the condensation of the silanol groups, this being important to conclude where the bonds Si–OH and Ln–OH are being more significant. The samples with the lowest percentage of mass loss in the third stage are those containing lanthanum and europium, these present a more resistant pore. The samples that showed a higher activation energy for the structural directional removal step are those with a more resistant silica wall. All samples showed surface acidity, showing that lanthanide oxides undergo changes in their basic character as a result of the influence of silica. The variation in pore diameter of Ln–SiO_2_ samples when compared to pure SiO_2_ is attributed to differences in bond strengths between Si–O–Si and Ln–O–Ln. The sample containing Eu presented the smallest pore diameter, evidencing that this sample presents the Eu–O bonds located more internally in the pores, presenting a thicker silica wall, while the materials containing the heavier lanthanides present a diameter of larger pore than the pure SiO_2_, these samples present a less resistant silica wall because they probably present a more significant contribution of the Ln–O–Ln bonds in the silica walls.

## Figures and Tables

**Figure 1 nanomaterials-13-00382-f001:**
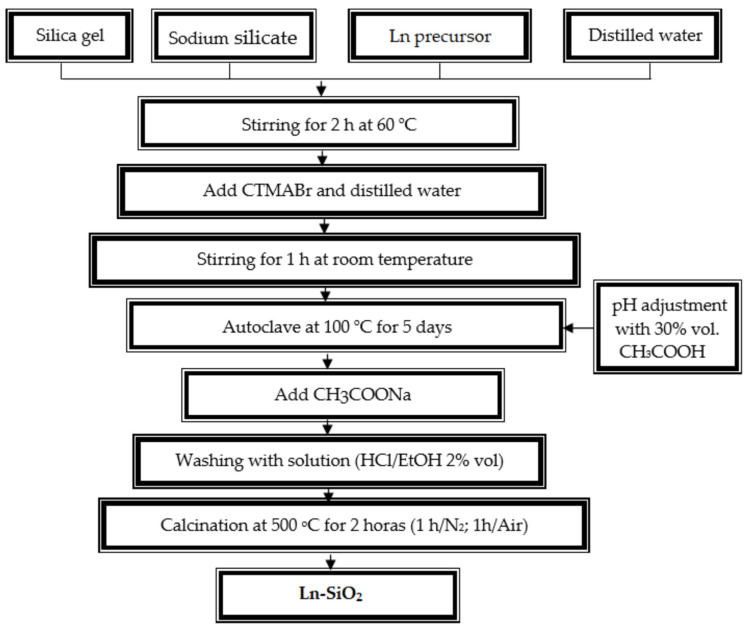
Flowchart showing the steps in the hydrothermal synthesis of Ln–SiO_2_ materials where Ln = La, Ce, Pr, Nd, Eu, Gd, Dy, Yb, Lu.

**Figure 2 nanomaterials-13-00382-f002:**
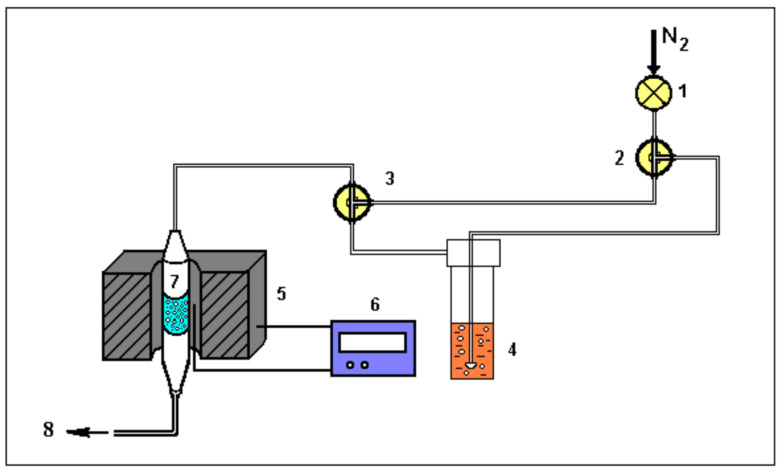
Schematic diagram used for acidity measurements, where: 1—valve for adjusting the N_2_ flow; 2 and 3—three-way valves; 4—saturator containing n-butylamine; 5—oven; 6—temperature controller; 7—reactor containing the sample; 8—exit of gases from the system.

**Figure 3 nanomaterials-13-00382-f003:**
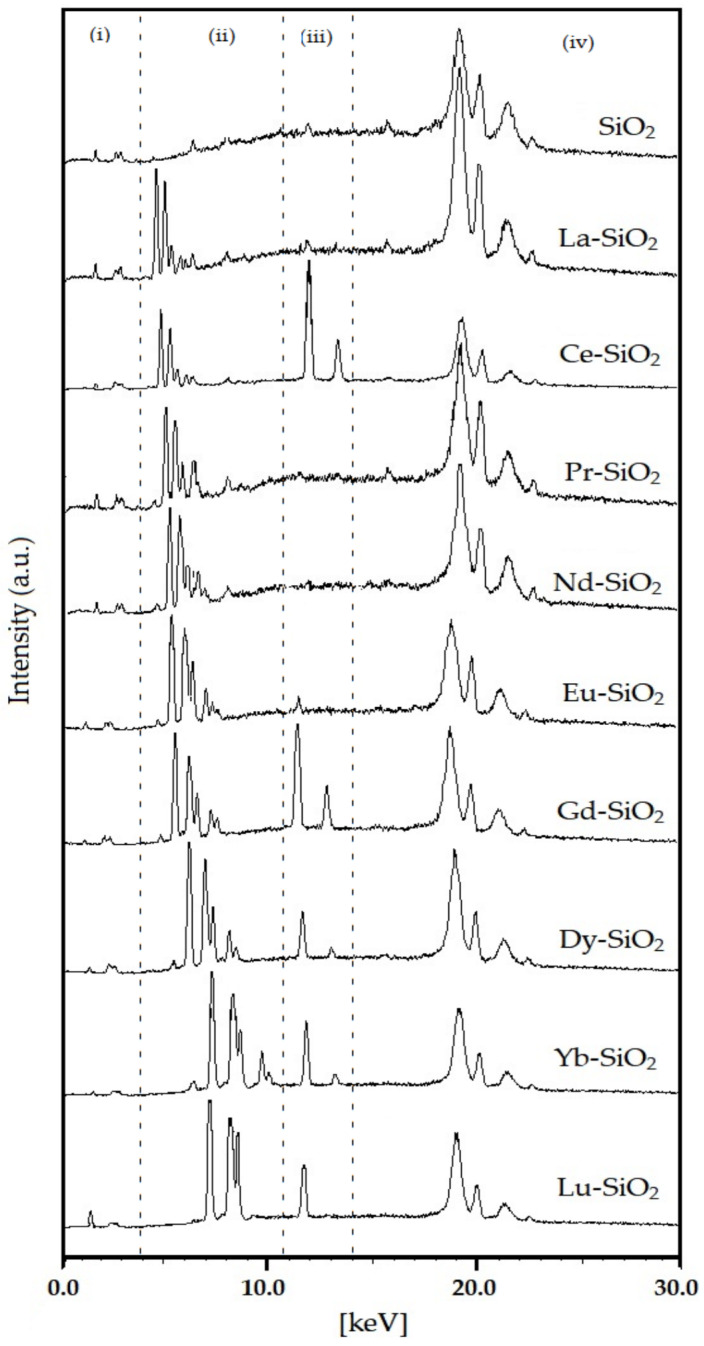
Energy dispersive XRF spectra of calcined samples Ln-SiO_2_ (Ln = La, Ce, Pr, Eu, Nd, Gd, Dy, Yb, Lu).

**Figure 4 nanomaterials-13-00382-f004:**
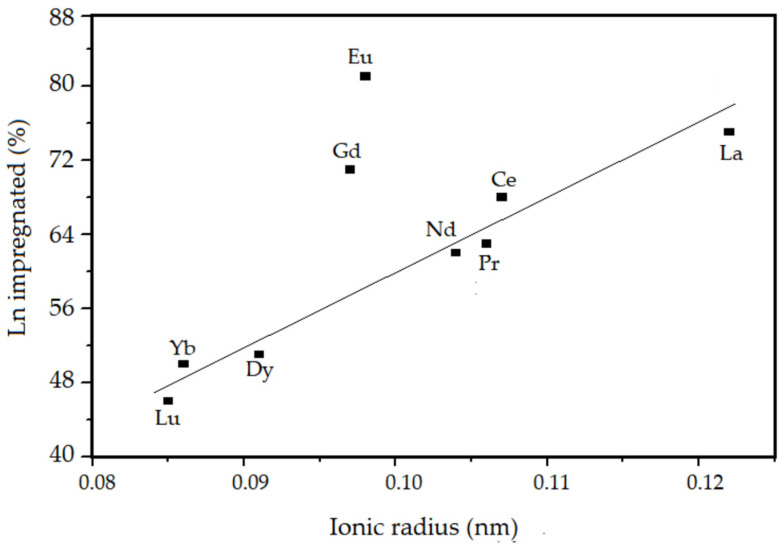
Lanthanide impregnated in the SiO_2_ as a function of ionic ratio of de Ln.

**Figure 5 nanomaterials-13-00382-f005:**
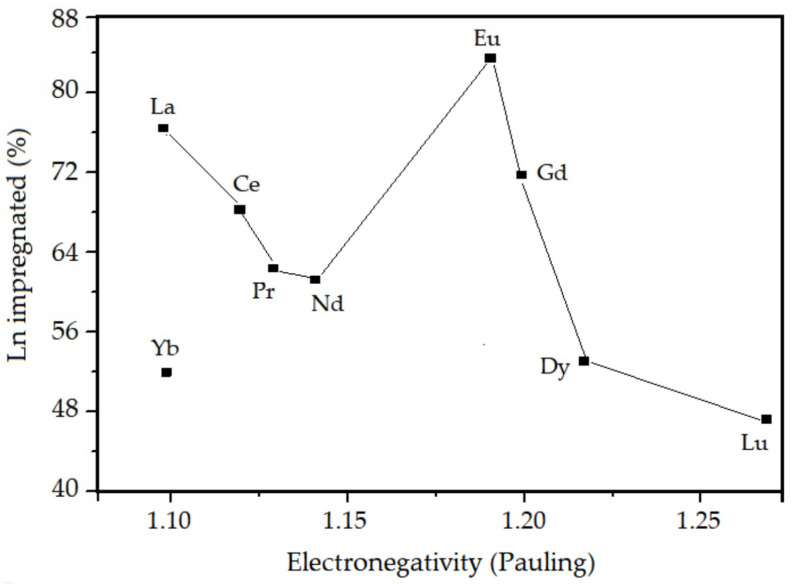
Lanthanide impregnated in the SiO_2_ as a function of electronegativity.

**Figure 6 nanomaterials-13-00382-f006:**
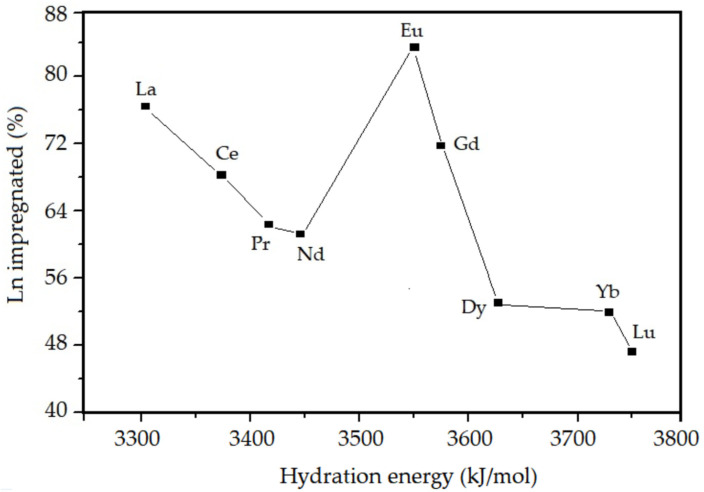
Lanthanide impregnated in the SiO_2_ as a function of hydration energy.

**Figure 7 nanomaterials-13-00382-f007:**
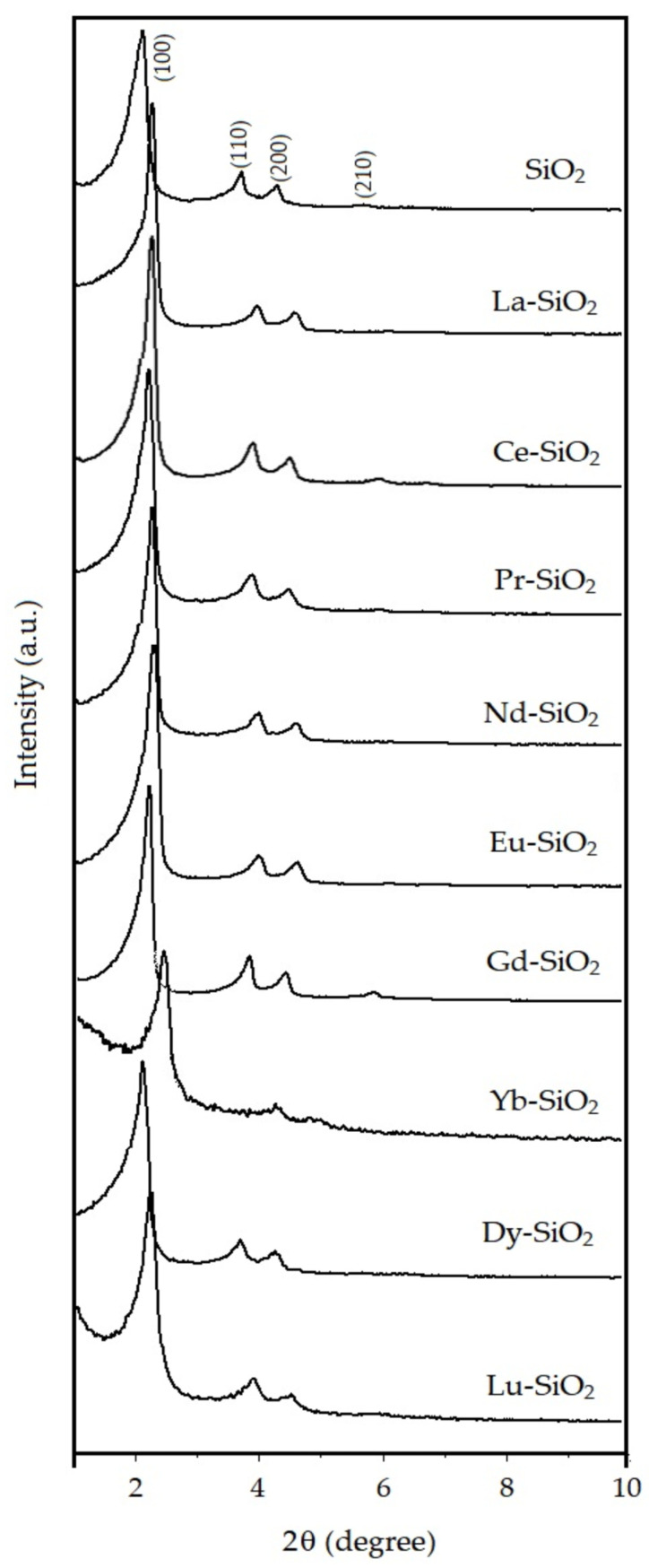
X-ray diffractograms of the calcined hexagonal SiO_2_ and Ln–SiO_2_.

**Figure 8 nanomaterials-13-00382-f008:**
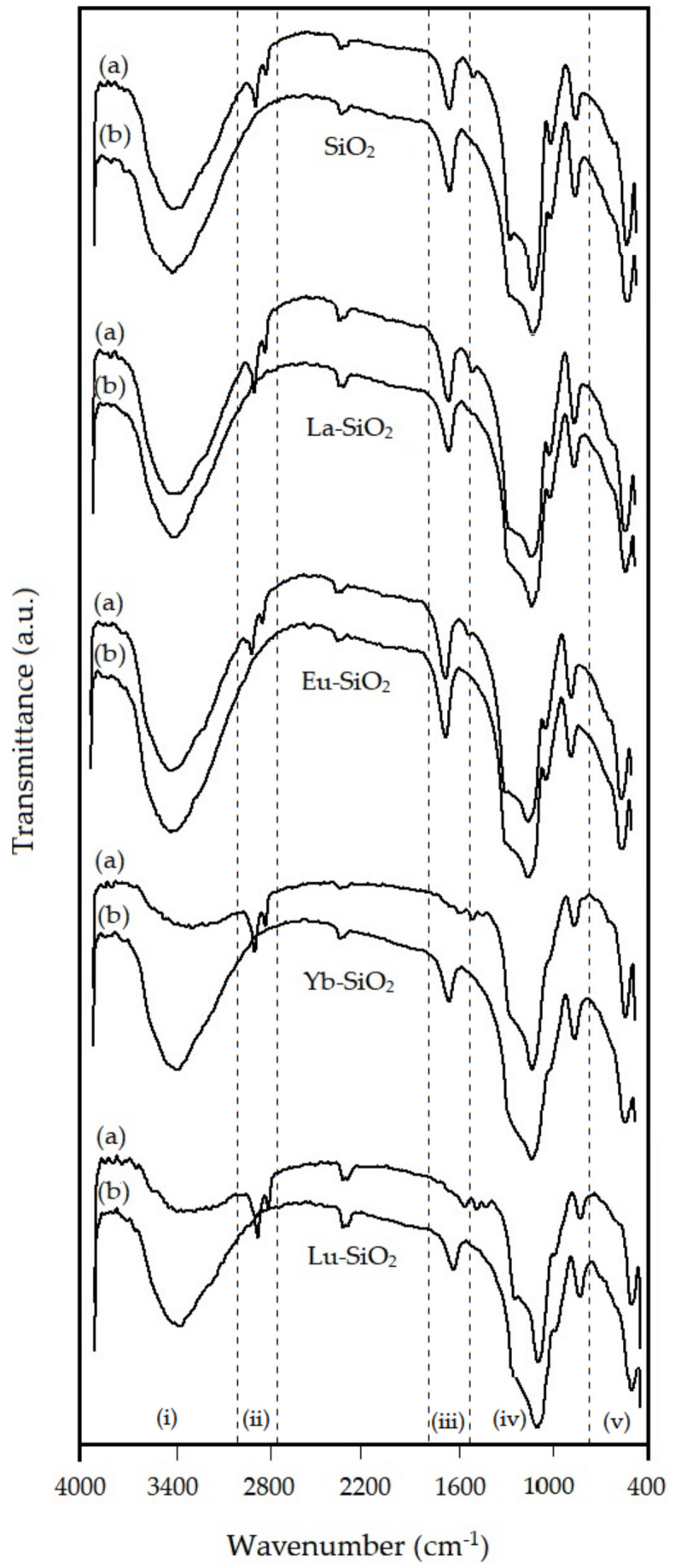
Fourier-Transform infrared spectra for SiO_2_ and Ln–SiO_2_: (a) before calcination and (b) calcined samples.

**Figure 9 nanomaterials-13-00382-f009:**
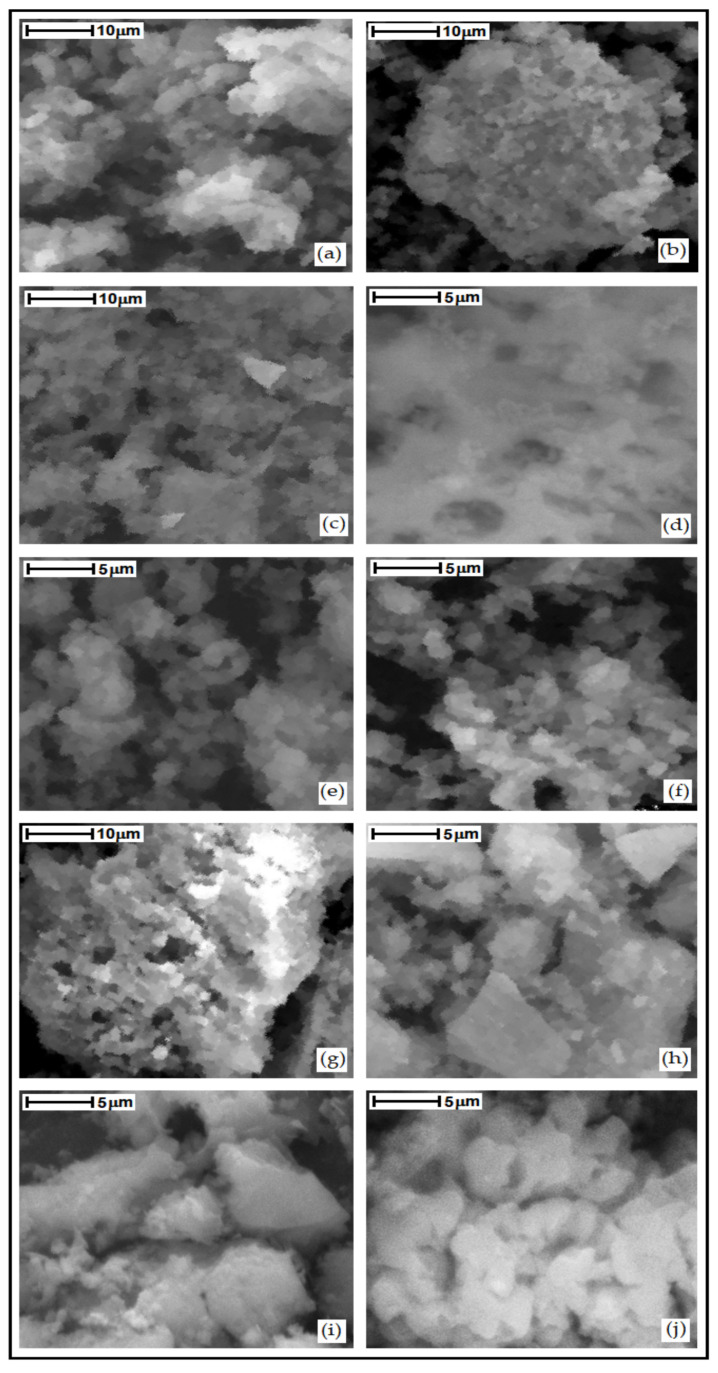
Scanning electron micrographies of the synthesized materials: (**a**) SiO_2_; (**b**) La–SiO_2_; (**c**) Ce–SiO_2_; (**d**) Pr–SiO_2_; (**e**) Nd–SiO_2_; (**f**) Eu–SiO_2_; (**g**) Gd–SiO_2_; (**h**) Dy–SiO_2_; (**i**) Yb–SiO_2_; (**j**) Lu–SiO_2_.

**Figure 10 nanomaterials-13-00382-f010:**
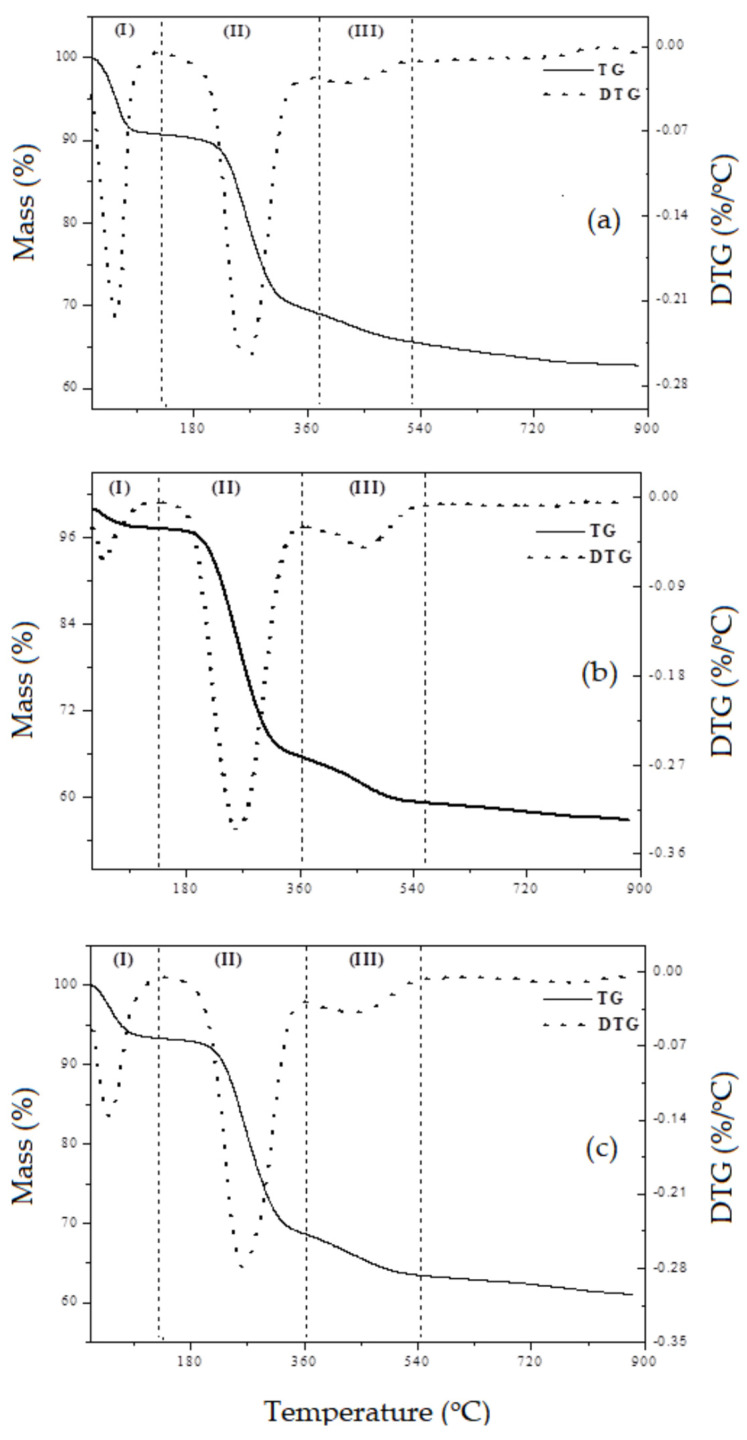
Thermogravimetry curves (TG/DTG) of the as synthesized materials: (**a**) SiO_2_; (**b**) Ce–SiO_2_ and (**c**) Lu–SiO_2_.

**Figure 11 nanomaterials-13-00382-f011:**
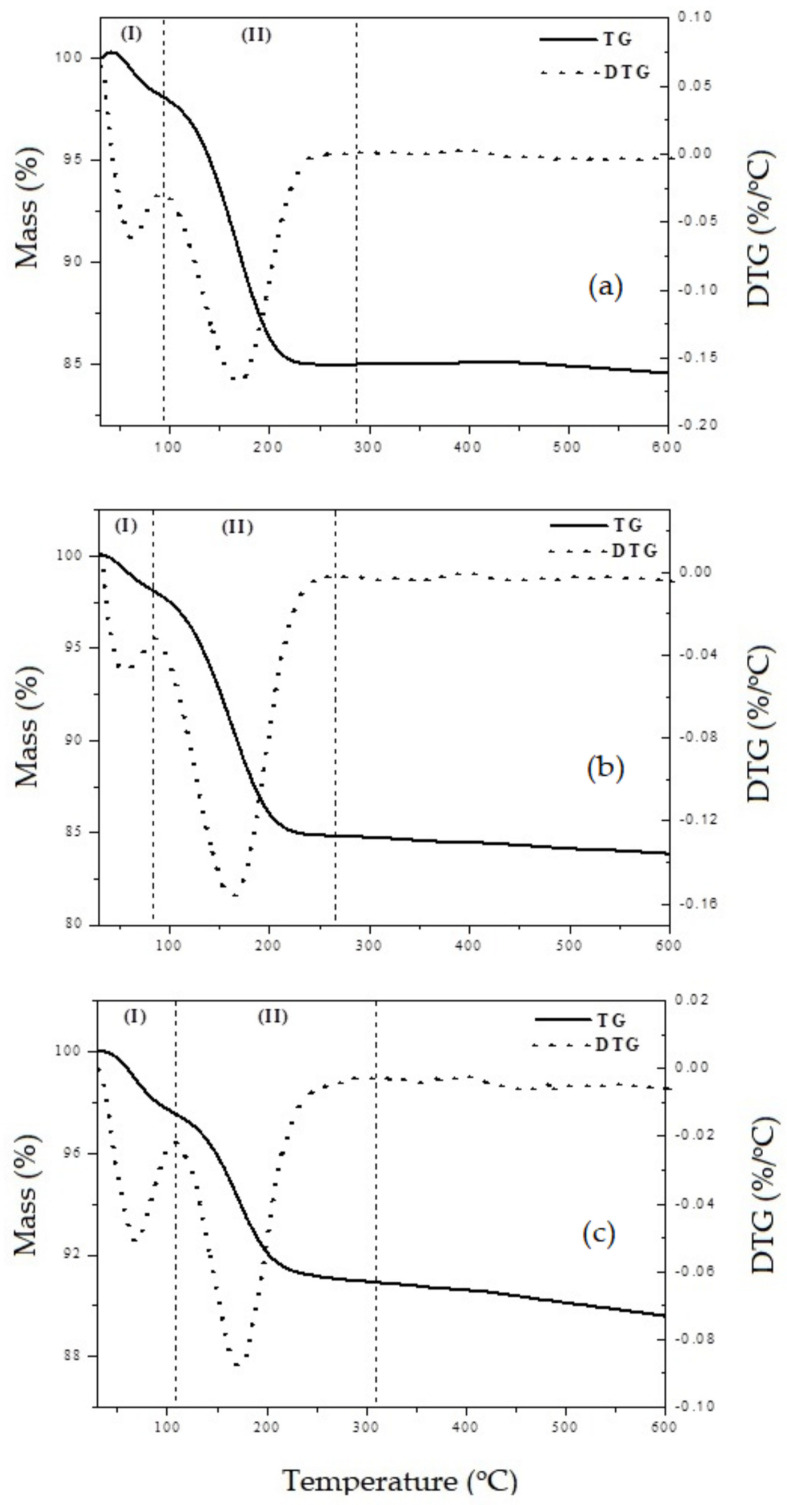
TG/DTG curves showing the thermal programmed desorption of n-butylamine from the adsorbed materials: (**a**) SiO_2_; (**b**) Ce–SiO_2_ and (**c**) Lu–SiO_2_.

**Figure 12 nanomaterials-13-00382-f012:**
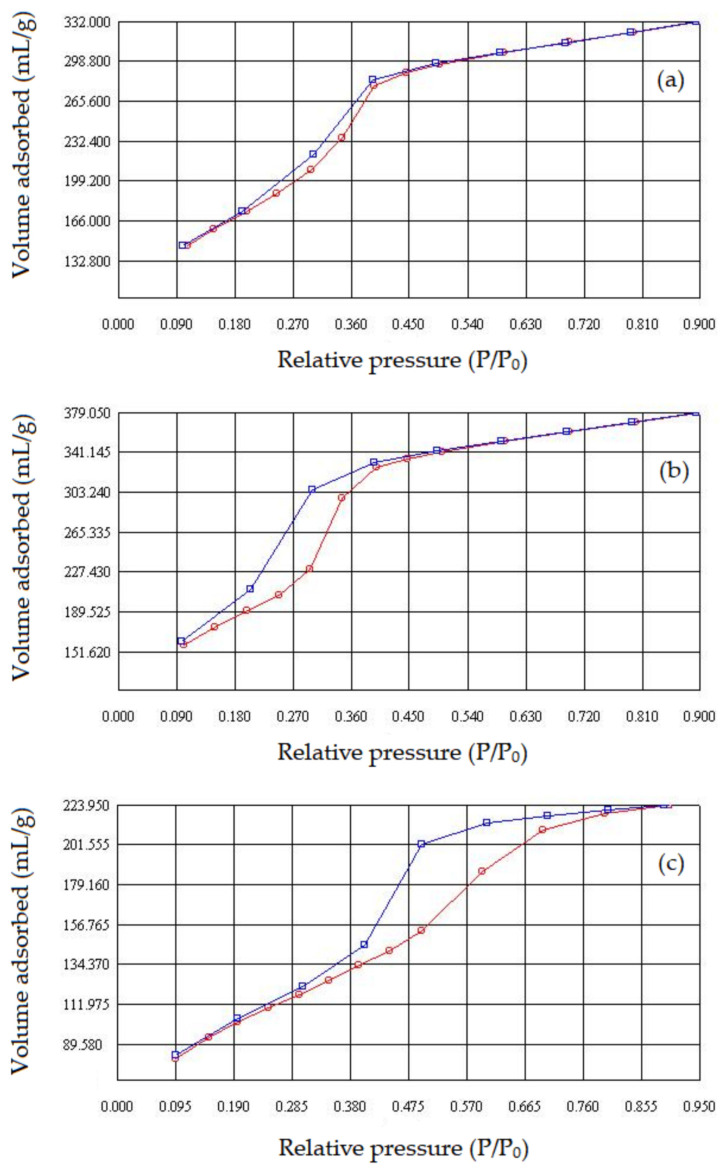
Nitrogen adsorption/desorption isotherms obtained at 77 K of calcined samples: (**a**) SiO_2_; (**b**) La–SiO_2_ and (**c**) Dy–SiO_2_.

**Table 1 nanomaterials-13-00382-t001:** Chemical composition of the SiO_2_ and Ln–SiO_2_, obtained from XRF spectra.

Sample	Chemical Composition of the Oxides (%)		Thoerical Molar Ratio Si/Ln	Experimental Molar Ratio Si/Ln	Amount of Ln Incorporated in the SiO_2_ (%)
	SiO_2_	Ln_2_O_3_			
SiO_2_	100	-	-	-	-
La–SiO_2_	95.52	4.48	50	68	75
Ce–SiO_2_	96.58	3.62	50	73	68
Pr–SiO_2_	96.58	3.42	50	80	63
Nd–SiO_2_	96.64	3.36	50	81	62
Eu–SiO_2_	95.47	4.53	50	62	81
Gd–SiO_2_	95.85	4.15	50	70	71
Dy–SiO_2_	96.94	3.06	50	98	51
Yb–SiO_2_	96.86	3.14	50	101	50
Lu–SiO_2_	97.04	2.96	50	109	46

**Table 2 nanomaterials-13-00382-t002:** Data obtained from X-ray diffration for calcined SiO_2_ and Ln–SiO_2_.

Sample	2θ (Degree)	(khl)	d (nm)	a_0_ (nm)
SiO_2_	2.19	(100)	4.03	4.65
La–SiO_2_	2.26	(100)	3.91	4.51
Ce–SiO_2_	2.26	(100)	3.91	4.51
Pr–SiO_2_	2.22	(100)	3.98	4.59
Nd–SiO_2_	2.28	(100)	3.88	4.48
Eu–SiO_2_	2.30	(100)	3.84	4.43
Gd–SiO_2_	2.26	(100)	3.91	4.51
Dy–SiO_2_	2.43	(100)	3.64	4.20
Yb–SiO_2_	2.09	(100)	3.94	4.55
Lu–SiO_2_	2.24	(100)	3.94	4.55

**Table 3 nanomaterials-13-00382-t003:** Thermogravimetric data for uncalcined SiO_2_ and Ln–SiO_2_ materials.

Sample	Temperature Range (°C)			Mass Loss (%)		
	(I)	(II)	(III)	(I)	(II)	(III)
SiO_2_	31–137	137–382	382–531	9.26	21.58	3.50
La–SiO_2_	31–136	146–353	353–464	7.36	9.23	1.79
Ce–SiO_2_	31–139	129–365	365–553	2.77	31.79	6.28
Pr–SiO_2_	31–137	137–357	357–448	2.11	15.69	3.53
Nd–SiO_2_	30–139	139–356	356–454	8.31	8.46	3.37
Eu–SiO_2_	30–137	137–345	345–494	7.51	5.98	1.99
Gd–SiO_2_	31–130	130–365	365–563	4.07	32.07	6.08
Dy–SiO_2_	31–141	148–352	352–492	4.25	8.96	4.84
Yb–SiO_2_	31–142	142–360	360–467	2.09	23.40	4.76
Lu–SiO_2_	30–142	142–368	368–540	6.72	24.67	4.94

**Table 4 nanomaterials-13-00382-t004:** Percentage of mass loss for the events associated with n-butylamine desorption from pure SiO_2_ and Ln–SiO_2_ materials.

Sample	Temperature Range (°C)		Mass Loss (%)		Acidity (mmol/g)
	I	II	I	II	
SiO_2_	30–93	93–286	1.9	13.0	2.2
La–SiO_2_	30–110	110–279	2.2	5.5	2.3
Ce–SiO_2_	30–83	83–266	1.9	13.3	2.3
Pr–SiO_2_	30–82	82–277	1.7	13.7	2.3
Nd–SiO_2_	30–97	97–269	2.8	10.1	1.6
Eu–SiO_2_	30–92	92–344	2.5	14.5	2.5
Gd–SiO_2_	31–91	91–274	1.5	9.8	1.5
Dy–SiO_2_	31–88	88–271	2.0	9.8	1.5
Yb–SiO_2_	31–7	77–276	1.1	10.1	1.6
Lu–SiO_2_	30–108	108–309	2.4	6.7	1.0

**Table 5 nanomaterials-13-00382-t005:** Structural properties of calcined materials of surface area, pore size diameter and silica wall thickness.

Sample	BET Surface Area (m^2^/g)	Pore Size Diameter (nm)	Silica Wall Thickness (nm)
SiO_2_	668	3.1	1.58
La–SiO_2_	732	3.2	1.31
Ce–SiO_2_	734	3.4	1.09
Pr–SiO_2_	699	3.4	1.23
Nd–SiO_2_	651	2.7	1.76
Eu–SiO_2_	721	2.6	1.83
Gd–SiO_2_	782	3.5	1.01
Dy–SiO_2_	376	3.7	0.52
Yb–SiO_2_	411	3.4	1.13
Lu–SiO_2_	412	3.7	0.90

## Data Availability

Not applicable.

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
