# Peer review of "Hydrothermal Synthesis and Properties of Nanostructured Silica Containing Lanthanide Type Ln–SiO2 (Ln = La, Ce, Pr, Nd, Eu, Gd, Dy, Yb, Lu)"

_nanomaterials, 2023, doi:10.3390/nano13030382_

Round 1
Reviewer 1 Report
In this manuscript, the authors have synthesized nanostructured lanthanide-silica materials of the Ln-SiO2 type (Ln = La, Ce, Pr, Nd, Eu, Gd, Dy, Yb, Lu) by the hydrothermal method, which was characterized by XRD, Fourier-Transform infrared absorption spectroscopy, scanning electron microscopy, TG, surface area by the BET method and acidity measurements to investigate the structure and properties. This work is meaningful, which can be published in Nanomaterials after a minor revision by addressing the following comments.
1 In 3.1. Chemical composition from XRF, the authors described that “The energy dispersive X-ray fluorescence (FRX) spectra of the catalysts are shown in Figures 3.” The FRX should be revised to XRF.
2 In 3.1. Chemical composition from XRF, the peaks of samples in the region (ii) of Figure 3 should be analyzed in detail.
3 In 3.3. Structural properties from FTIR, the authors described that “The spectra were carried out before and after calcination, as noted in the figure xxx, as letters (a) and (b), respectively.”. The figure xxx should be revised.
4 In 2.2. Hydrothermal synthesis procedures, the calcination process in Figure 1 should be described in detail.
5 The number of Figure was wrong and should be revised.
6 The relevant literatures such as Materials Reports: Energy 1 (2021) 100018 should be cited.
Author Response
Response to Review 1:
1) In 3.1. Chemical composition from XRF, the authors described that “The energy dispersive X-ray fluorescence (FRX) spectra of the catalysts are shown in Figures 3.” The FRX should be revised to XRF.
- Revised.
2) In 3.1. Chemical composition from XRF, the peaks of samples in the region (ii) of Figure 3 should be analyzed in detail.
The following sentence was included in the manuscript, with references:
“The compounds La-SiO2, Pr-SiO2, Nd-SiO2 and Eu-SiO2, only peaks in the region (ii) were observed. The presence of peaks in region (ii) are probably due to a high degree of covalenty coordination of the Ln-O-Si bonds (Ln = La, Pr, Nd and Eu) in the materials, suggesting the presence of trivalent cations of the lanthanides [43,44].”
[43] Gregory R. Choppin, G.R. Covalency in f-element bonds. Journal of Alloys and Compounds, 2002, 344, 55–59.
[44] Bagusa, P.S.; Nelin, C.J. Covalent interactions in oxides. Journal of Electron Spectroscopy and Related Phenomena, 2014, 194, 37–44
3) In 3.3. Structural properties from FTIR, the authors described that “The spectra were carried out before and after calcination, as noted in the figure xxx, as letters (a) and (b), respectively.”. The figure xxx should be revised.
- Revised.
4) In 2.2. Hydrothermal synthesis procedures, the calcination process in Figure 1 should be described in detail.
- The following sentence was included in the manuscript, as one paragraph:
"According to Figure 1, the last step to obtain the nanoporous materials is the calcination. The calcination process of the series of the SiO2 and Ln-SiO2 nanomaterials occurred in two steps where initially the samples were subjected to a heating hate of 10 oC/min from room temperature to 500 oC in inert nitrogen atmosphere flowing at 100 mL/min. After reaching the temperature of 500 oC, the system remained under this condition for 1 hour. Then, the nitrogen flow was changed for synthetic air at the same flow rate for an additional time of 1 hour. This calcination process aims to remove the structural template from the pores of the materials."
5) The number of Figure was wrong and should be revised.
- Revised
6) The relevant literatures such as Materials Reports: Energy 1 (2021) 100018 should be cited.
- The suggested refference was cited:
[42] Belotti, A.; Liu, J.; Curcio, A.; Wang, J.; Wang, Z.; Quattrocchi, E.; Effat, M.B.; Ciucci, F. Introducing Ag in Ba0.9La0.1FeO3-δ: Combining cationic substitution with metal particle decoration. Materials Reports: Energy, 2021, 1, 100018
Reviewer 2 Report
The study reports an investigation on the properties of lanthanide doped mesoporous silica prepared in sol gel process. Various experimental techniques were employed and the results showed certain interesting tendencies on the effect of the different lanthanides. The investigations seem to be performed accurately. The paper contains novel ideas and data, and is of interest for readers of Nanomaterials.
Minor corrections are necessary.
1. Grammar should be checked and corrected throughout the text. For example:
- line 49 “… one of the most important materials developed so far with … “
- line 51
- line 253 – please check the wavelength and the units.
- line 254 – mistake in units
- line 282
2. The experimental data of XRD, thermal analysis or nitrogen sorption, although well presented and interpreted are missing a comparison to literature data on pure or metal-containing MCM-41 materials.
Thus, a few papers can be recommended to complete this gap, such as these for the TG part.
10.1007/s10973-011-1908-8
10.1016/S0040-6031(00)00637-7
10.3390/gels8070443
Author Response
Response to Reviewer 2
Thanks for the comments and review.
1. Grammar should be checked and corrected throughout the text. For example:
- line 49 “… one of the most important materials developed so far with … “
- line 51
- line 253 – please check the wavelength and the units.
- line 254 – mistake in units
- line 282
- They are corrected. All corrections are marked in red color.
2. The experimental data of XRD, thermal analysis or nitrogen sorption, although well presented and interpreted are missing a comparison to literature data on pure or metal-containing MCM-41 materials.
Thus, a few papers can be recommended to complete this gap, such as these for the TG part.
10.1007/s10973-011-1908-8
10.1016/S0040-6031(00)00637-7
10.3390/gels8070443
- The suggested references were cited in the manuscript.
